# Optimization of Ultrasound Assisted Extraction (UAE) of Kinsenoside Compound from *Anoectochilus*
*roxburghii* (Wall.) Lindl by Response Surface Methodology (RSM)

**DOI:** 10.3390/molecules25010193

**Published:** 2020-01-02

**Authors:** Biyun Yang, Mengyuan Zhang, Haiyong Weng, Yong Xu, Lihui Zeng

**Affiliations:** 1College of Horticulture, Fujian Agriculture and Forestry University, Fuzhou 350002, China; yangbiyun2010@126.com (B.Y.); zy15659121197@126.com (M.Z.); 2College of Mechanical and Electronic Engineering, Fujian Agriculture and Forestry University, Fuzhou 350002, China; hyweng@fafu.edu.cn; 3Institute of Machine Learning and Intelligent Science, Fujian University of Technology, 33 Xuefu South Road, Fuzhou 350118, China

**Keywords:** kinsenoside, *Anoectochilus roxburghii*, UAE, UPLC, RSM

## Abstract

The purpose of this study was to establish an extraction method for the kinsenoside compound from the whole plant *Anoectochilus roxburghii*. Ultrasound assisted extraction (UAE) and Ultra-high performance liquid chromatography (UPLC) method were used to extract and determine the content of kinsenoside, while response surface method (RSM) was used to optimize the extraction process. The best possible range for methanol concentration (0–100%), the liquid-solid ratio (5:1–30:1 mL/g), ultrasonic power (240–540 W), duration of ultrasound (10–50 min), ultrasonic temperature (10–60 °C), and the number of extractions (1–4) were obtained according to the single factor experiments. Then, using the Box-Behnken design (BBD) of response surface analysis, the optimum extraction conditions were obtained with 16.33% methanol concentration, the liquid-solid ratio of 10.83:1 mL/g and 35.00 °C ultrasonic temperature. Under these conditions, kinsenoside extraction yield reached 32.24% dry weight. The best conditions were applied to determine the kinsenoside content in seven different cultivation ages in *Anoectochilus roxburghii*.

## 1. Introduction

*Anoectochilus roxburghii* (hereafter shortened as *A. roxburghii*), a traditional herb used in China for medicinal and culinary purposes, grows in the wet place under the evergreen broadleaved or bamboo forest, which contains several bioactive compounds, such as kinsenoside, polysaccharides, flavonoids, and glycosides [1,2,3].

Among those, kinsenoside (*3*-*O*-β-*D*-glucopyranosyl-(*3R*)-hydroxy butanolide) is a major compound of *A. roxburghii* [1,3]. Ito et al. [4] first reported the isolation and structural characterization of it, which exists in *Anoectochilus koshunensis* and *Anoectochilus formosanus.* Researchers have found that kinsenoside had a different multiple pharmacologic activity compared with polysaccharide and flavonoids [5,6,7,8]. As a potential immunosuppressive drug for autoimmune hepatitis, it has a vascular protective effect under high glucose conditions and has been found to enhance the oxidative resistance of diabetic mice [9,10,11]. In addition, it has a reparative effect on damaged insulin cells and some other functions, such as anti-hyperliposis, anti-hyperglycemia, anti-osteoporosis, etc. [12,13,14,15]. As the extracted active compounds from *A. roxburghii,* kinsenoside could be used as the ingredients for health care medicine and foods.

Many methods have been used to extract the compounds from *A. roxburghii*, such as Soxhlet extraction, microwave-assisted extraction (MAE), and supercritical fluid extraction (SFE) [16,17]. As the extraction process via these techniques is generally time-consuming, operating cumbersome with a relatively high cost, the ultrasound assisted extraction (UAE) method is generally used which promotes mass transfer and rupture of cell walls through acoustic cavitation, improving the efficiency and optimizing the extraction yield [18,19,20]. In addition, it is considered to be an emerging potential technology in green food processing techniques [21]. Since it has received considerable attention due to its beneficial properties and green impacts, including preserving their structural and molecular properties, high extraction efficiency, high reproducibility, short extraction time and low solvent consumption, easy operation, low cost, and low pollution to environment [20,21,22,23,24]. As such, UAE method has been widely used to extract kinsenoside [2,25,26], but the extraction process has not been optimized.

As for the experimental method, the response surface methodology (RSM) is a very helpful strategy to maximize the yields of the compounds from plants by optimizing operational factors [22,27], and the Box-Behnken design (BBD) is a useful tool to perform such multi-factor experiments and allows obtaining a predictive model [28]. The most advantageous feature of the RSM is that it reduces the number of experimental runs, saving energy, time, and raw materials as well as improving the quality of the information obtained from the results compared to the individual study of each variable. It is widely used for optimizing the extraction of polysaccharides, oil, flavonoids, phenolic compounds, anthocyanins, soluble sugars, proteins, and polyphenols from different plant materials [18,29,30,31,32,33,34,35].

The main objectives of this study were to optimize UAE for the extraction of kinsenoside from plant *A. roxburghii* using the RSM. 

The methods for the kinsenoside yield extraction (%) under the optimal UAE conditions were used for the extraction from *A. roxburghii* at different cultivation age. Among others, kinsenoside have been categorized under nutraceutically active compound and their role has been justified in treating various ailments. Thus, the present study was designed to determine the optimum UAE condition for maximizing kinsenoside extraction yield from *A. roxburghii*.

## 2. Results and Discussions

Before optimizing the extraction method using the Box-Behnken-RSM methodology, a methanol extract was prepared using the method reported by Zhang et al. [36] and analyzed by UPLC-MS to determine the content of kinsenoside compound as reported by Luo et al. [25]. Accordingly, 20% of methanol concentration, the liquid-solid ratio of 15:1 mL/g, 40 °C of extraction temperature was selected as the central point in BBD.

### 2.1. Selection of Factors and Their Levels by Single Factor Analysis

Figure 1 shows the results of the ultrasound assisted extraction at different methanol concentrations, liquid-solid ratios, ultrasonic powers, ultrasonic temperatures, duration of ultrasounds, and the number of extractions. 

#### 2.1.1. Methanol Concentration

It can be observed from Figure 1A that in terms of the kinsenoside extraction yield, the optimal methanol concentration is 20%. This showed that a suitable mixture of methanol and water can facilitate the extraction of kinsenoside compound that is best dissolved in the solvent at a suitable ratio. The extraction effect of this mixed solvent for kinsenoside compound is better than that by the methanol alone [25,36] or the distilled water alone [37].

#### 2.1.2. The Liquid-Solid Ratio

Kinsenoside yield experienced a growth with the increase of the liquid-solid ratio from 5:1 to 15:1 mL/g, but a further increase in the liquid-solid ratio led to a decline in the extraction yield. This result is in good agreement with previous research [38,39]. The liquid-solid ratio plays a significant role in the completeness of the extraction. Generally, a higher concentration gradient allows a faster rate of mass transfer from the solid to the solvent. However, this effect is attenuated if the liquid-solid ratio is too high, since the main mass transfer is limited in the range within the liquid-solid ratio to avoid solvent waste and extraction cost [39,40]. Therefore, based on this observation, the optimal liquid-solid ratio was selected to be 15:1 mL/g for further investigation. This also demonstrates that the extraction efficiency is mainly dominated by the dissolving capability of the solvents.

#### 2.1.3. Ultrasonic Temperature

With the increase in ultrasonic temperature from 10 to 60 °C, a significant linear increase in extraction yield was recorded, with the maximum extraction yield at 40 °C (Figure 1C). The result was inconsistent with the studies reported by other researchers in which a temperature of 25 °C or 100 °C [37] were used for the extraction of kinsenoside from *A. roxburghii*. A higher temperature during the ultrasonication would provide a higher kinetic energy to loosen the sample cell structure in order to greatly intensify mass transfer phenomena, which lead to the greater rate of diffusion [32,41]. However, a higher temperature in the medium could increase the vapor pressure which affects the intensity of acoustic cavitation and reduces the surface tension of the solvent. A too high temperature exceeding 40 °C may destroy the cell tissues and result in a degradation of the product [39]. Therefore, it is beneficial to optimize the extraction temperature in order to have a better extraction yield and higher product quality.

#### 2.1.4. Duration of Ultrasound

Increasing the duration of ultrasound from 10 min to 20 min, the extraction yield was significantly increased (Figure 1D). However, further increase in ultrasonic time significantly decreased the extraction yield. As reported by Tomšik et al. [20], an increase in extraction time provides longer contact of the compound with solvent and facilitates a better diffusion of the target compounds. However, a long-duration extraction is not suitable for bioactive compounds, likely because of the degradation of the product [31]. In this work, the extraction time was chosen to be 20 min, which is comparatively appropriate by results obtained elsewhere [31,39].

#### 2.1.5. Ultrasonic Power

As illustrated in Figure 1E, the rise of ultrasonic power resulted in increasing the kinsenoside yield. This increasing trend lasted up to 360 W. After that, with a further increase of the ultrasonic power, kinsenoside yield is reduced. The results showed that when the ultrasonic power is too high, it may accelerate the drift of the extraction and would make the bubbles in the solvent produce more rapidly, causing both the wall breaking effect and the dissolution rate of kinsenoside decreased [42,43]. Therefore, 360W was considered the optimal UAE power due to the high kinsenoside yield and the relatively low energy consumption.

#### 2.1.6. Number of Extractions

With the increase of the number of extractions, the kinsenoside also increases accordingly. The yield of kinsenoside reached 18.70% DW for the first extraction, 33.20% DW for the second, 36.30% DW, and 39.10% DW for the third and the fourth extraction, respectively. The results showed that when the number of extractions is greater than two, the small amount of the extract increased would not match the energy and time consumed during the process. Hence, two times of extraction is considered the best and used in our experiments.

### 2.2. Model Fitting

The optimal kinsenoside extraction conditions were determined by analysis using Design Expert version 8.0.5.0 software (Stat-ease INC., Minneapolis, MN, USA). Summarizing the results are shown above, the resulting model is expressed as:*y* = 30.31 − 3.11A − 1.69B − 2.26C − 0.082AB + 1.95AC + 0.94BC − 7.64A^2^ − 3.80B^2^ − 0.038C^2^.(1)

This model was built by fitting the second-order polynomial model of the coded values to the experimental values. It can be concluded that the models derived for kinsenoside were found significant, confirming that the model could adequately fit the experimental data [44,45]. The magnitude of each coefficient in this equation directly reflects the degree of influence of each factor on the index value, and the sign of it reflects the direction of influence.

In order to investigate the goodness of fit and the adequacy of the model for the responses, the analysis of variance (ANOVA) method was performed. The results showed that the effect of primary terms A, B, C, cross terms AC, and quadratic terms A^2^ and B^2^ on the yield was extremely significant. This indicates that the test factor is not a simple linear relation to the response value and the quadratic term has a great influence on the responses. As it can be observed, the statistical correlation coefficient (R^2^) is higher than 0.9, which implicates the adequacy for prediction of the experimental results. Furthermore, Adj R-Squared (Radj^2^) was close to R^2^, implying reasonable adjustment of the model to the experimental data [46]. The “Lack of Fit F-value” of 0.13 implies the lack-of-fit was not significant relative to the pure error, indicating that the experimental error is very small and the suitability of the model to accurately predict variation [43,47]. The F value of the model was 117.59, which indicates that the model is extremely significant with a probability of 0.01%. The coefficient of variation (CV = 2.71%) is less than 5%, indicating that the model has good reproducibility. This showed that the regression model was reasonable and was deemed adequate to predict the yield of kinsenoside.

### 2.3. Effect of UAE Factors on Kinsenoside Extract Yield

Based on Equation (1) and the statistical analysis, the regression model of the yield of kinsenoside was visualized using the three-dimensional response surface diagrams and contour maps (Figure 2a–f). Figure 2 illustrates the response surface for the dependent variable (yield) as a function of two factors when the third was sustained and kept constant. The results showed that three factors had an interaction, and the interaction between methanol concentration and ultrasonic temperature was the most significant, followed by the interaction between other factors.

The interactive effect of ultrasonic temperature (A) and methanol concentration (B) on the extraction yield of kinsenoside from *A. roxburghii* at a fixed ratio of liquid-solid (15 mL/g) is depicted as surface and contour plots in Figure 2a,b, respectively. The mutual interaction between ultrasonic temperature (A) and methanol concentration (B)was not significant in improving the yield of kinsenoside due to large *p*-values (>0.05) (Table 1). The negative coefficient in the model term (−0.082AB) (Equation (1)) reveals antagonistic behavior between the variables. Figure 2c,d illustrate the surface and contour plots for the interactive effect between ultrasonic temperature (A) and the liquid-solid ratio (C) on extraction yield of kinsenoside from *A. roxburghii* at a fixed methanol concentration (20%). Notably, the mutual interaction between ultrasonic temperature and the liquid-solid ratio was highly significant (*p* = 0.007) (Table 1). The interactive term AC (+1.95AC) was suggestive of a positive effect between ultrasonic temperature (A) and the liquid-solid ratio (C) (Equation (1)). The surface and contour plots for the interactive effect of methanol concentration (B) versus the liquid-solid ratio (C) on the extraction yield of kinsenoside from *A. roxburghii* at a fixed ultrasonic temperature (40 °C) is illustrated in Figure 2e,f. The interactions of the two variables were significant due to a small *p*-value (0.0268) (Table 1). The interaction term (+0.94BC) between methanol concentration (B) and the liquid-solid ratio (C) (Equation (1)) indicated a synergistic effect between the two variables.

Based on the analyses above, the optimum extraction process conditions are: the ultrasonic temperature is 35 °C, the methanol concentration is 16.33%, and the liquid-solid ratio is 10.83:1 mL/g. The maximum predicted yield under these conditions was 33.00% DW. Finally, according to the optimized extraction process obtained from the response surface test, a verification test was carried out. The results showed that the average yield of kinsenoside was 32.24% DW, and the relative error with the predicted value (33.00% DW) was only 2.30%. There are few studies about the optimization method for the extraction of kinsenoside, but our result can be compared with other similar methods for the extraction of kinsenoside. Jin et al. [2] used continuous immersion bioreactor systems to investigate several factors affecting rhizome biomass and bioactive compound accumulation and the maximum kinsenoside accumulation (32.54% DW) was determined when a bioreactor was inoculated with 12.5 g/L (fresh weight) of rhizomes. Luo et al. [13] added salicylic acid (SA) and methyl jasmonate (MeJA) to the culture medium of rhizome suspension culture, and the maximum kinsensode accumulation (661.4 mg/g DW) was investigated under the SA-treated group. They adopted UAE method along with HPLC-MS to determine the content of kinsenoside. However, the extraction process parameters were not optimized. Our experiment has optimized the key factors for the extraction of kinsenoside, which is expected to be of certain reference significance for later research experiments.

### 2.4. Evaluation of the Content of Kinsenoside at Different Cultivation Age

In this study, the contents of kinsenoside in *A. roxburghii* at seven different cultivation ages were analyzed. As demonstrated in Figure 3, the content of kinsenoside in *A. roxburghii* was different at different ages with the highest at the tissue-culture stage. The plant hormones (such as 6-BA, NAA) added to tissue culture medium possibly played a positive role in promoting the accumulation of kinsenoside. After transplanting planting site under the forest, the content of kinsenoside radically decreases, especially in the first and second cultivating months. The reason for this may be that the metabolism rate in the plant is accelerated and nutritional ingredients (such as kinsenoside, polysaccharide, phenolic) which accumulated in the tissue-culture stage are consumed for the plant to adapt to the new environment of cultivation. From the second to fifth cultivating months, the content of kinsenoside increases as the cultivating time increases. This indicates that the energy generated due to photosynthesis is greater than the energy consumed by metabolism when the plants adapted to the new growth environment in this stage. However, kinsenoside content decreases as the cultivation month further increases. Since *A. roxburghii* is a perennial herb, the secondary metabolites may depend on different cultivation age for the development of the plant. These results agree with those of previous workers who reported that at different cultivation ages, the contents of bioactive compounds varied in *A. roxburghii* [48]. Chen et al. [48] found that the peaks of kaolinite started to appear from the sixth cultivated month of the *A. roxburghii* by FT-IR spectroscopy, which proved that as the cultivation month increased, more inorganic minerals were absorbed from the soil. The results in this study indicated that tissue-cultured *A. roxburghii* is proposed to be suitable raw materials in the pharmaceutical industry for enhancing the kinsenoside productivity for medicinal, health care, food, and cosmetic products.

## 3. Materials and Methods

### 3.1. Plant Material

The samples (*A. roxburghii* ‘Gongxiang No.2’) were offered by Fujian Gongxiang Eco-agriculture Technology Co. Ltd., planted in September 2017 and harvested from November 2017 to April 2018 at Yukeng Village (Fujian, China, 25°48′37.08″ N, 119°10′5.92″ E). *A. roxburghii* plants for the establishment of the optimum extraction conditions were cultivated under a forest for seven months. After being washed thoroughly, the whole plant of the *A. roxburghii* was dried by a freeze-drying machine (LGJ-S24, Beijing Four-Ring Scientific Instrument Factory Co. Ltd., Beijing, China) and then crushed to an average particle size of 0.4 mm with a grinder. The *A. roxburghii* medicinal powder was obtained and stored in a refrigerator at −40 °C for use.

### 3.2. Chemicals and Kinsenoside Standard

Methanol and acetonitrile (HPLC grade) were purchased from Merck (Darmstadt, Germany) and the pure water from Hangzhou Wahaha group Co. Ltd. (Hangzhou, China). The standards of kinsenoside (≥98% assay) were supplied by Shanghai Lichen Biotechnology Co. Ltd. (Shanghai, China).

The linear regression equation of the sample was obtained as y = 84911x + 192050 after using the concentration of the standard solution as *x* axis and the corresponding relative peak area as *y* axis. The correlation coefficient is 0.995. The linear range is 5 µg–500 µg, which meets the need of quantitative analysis. The UPLC chromatograms of kinsenoside reference substances was shown in Figure 4.

### 3.3. Selection of Variables

The effects of different variables, such as solvents type and concentration, duration of ultrasound, temperature, sample-to-solvent ratio, number of extractions, etc. are known to affect extraction yield [20,49]. In this context, six processing variables, i.e., methanol concentration (%), the liquid-solid ratio (mL/g), ultrasonic power (%), ultrasonic temperature (°C), duration of ultrasound (min), and the number of extractions were tested in a preliminary experiment to investigate and determine the effects of the factors and levels of the kinsenoside extraction process. The selected six independent variables were investigated at six levels (shown in Table 2). All the experiments were carried out in triplicates. Selection of the best extraction process was based on the maximum yield of kinsenoside.

### 3.4. Experimental Design for the Kinsenoside Extraction

Box-Behnken design (BBD) experiment was applied to determine the extraction variables for the kinsenoside compound from *A. roxburghii*. A three level (lower (−1), middle (0) and higher (+1)), three-factor strategy was applied for the design of experiments, model building, and data interpretation (the experimental design matrix is shown in Table 3). The independent factors selected for the optimization of ultrasound assisted extraction (UAE) include the ultrasonic temperature (°C) (A), methanol concentration (%) (B), and the liquid-solid ratio (mL/g) (C). A total of 17 experimental runs were used to normalize parameters and to evaluate the effect on the kinsenoside compound. Table 4 shows the experimental design (coded and natural values of the factors) for each run.

The content of kinsenoside was considered as the response factor and the second order polynomial equation was determined as:(2)$$y=\beta_{0}+\sum_{i=1}^{3}\beta_{i}x_{i}+\sum_{i=1}^{3}\beta_{ii}x_{i}{}^{2}+\sum_{i=1}^{3}\sum_{j>i}^{3}\beta_{ij}x_{i}x_{j},$$
where *y* is the response variable, *β*_0_ the intercept, *β_i_* the linear regression coefficient for *i*th factor, *β_ii_* the coefficient for quadric, and *β_ij_* for the cross-product term. *x_i_* and *x_j_* are the independent variables.

To test the predicted model on the response variable, an analysis of variance (ANOVA) with 95% confidence level was performed to evaluate the effect of each factor. Besides, the regression coefficient (R^2^), the *p*-value of the regression model and the *p*-value of the lack of fit (LOF) were employed to evaluate the fitness of the regression model. The three-dimensional response surface plots were generated for each response by keeping one response variable at its optimal value and plotting the other two independent factors with the response.

### 3.5. Determination of Kinsenoside Compound of A. roxburghii

The powdered sample was extracted according to the method described by Zhang et al. [36] with some modifications. *A. roxburghii* powder (0.1 g) at the ratio 20:1 mL/g of liquid to solid was extracted using methanol (2 mL) at 30 °C for 45 min on 420 W ultrasonic power bath (KQ-600DE CNC ultrasonic cleaner, Kunshan Ultrasonic Instrument Co. Ltd., Suzhou, China). After centrifugation at 10,000 rpm for 10 min, the supernatant was drained to a new tube. Then, the ultrasonic power bath and centrifugation were conducted again, and the supernatant was drained to the former tube to form 2 mL of supernatant (if the supernatant is less than 2 mL, add some methanol solution until it reaches 2 mL). All samples were filtered through a 0.22 µm PVDF filter (Millipore, Burlington, MA, USA) and stored in a refrigerator at −20 °C for the use of analyses.

The content of the kinsenoside compound was determined by UPLC in accordance with the method described by Luo et al. [25] with some modifications. Kinsenoside in extract preparations was determined by a Waters Acquity UPLC system equipped with a UV detector (215 nm). The separation of kinsenoside was performed on a Waters Acquity UPLC column (C18, 1.7 μm, 2.1*100 mm) under the following conditions: columns temperature 30 °C, injection volume 5 µL, mobile phase, the ratio of comprising water, acetonitrile and methanol 90:5:5 (*v*/*v*/*v*), and flow rate 0.2 mL/min. All analyses were repeated three times.

### 3.6. Evaluation of the Content of Kinsenoside at Different Growing Stage

The samples at different growing stage were obtained from November 2017 to April 2018. *A. roxburghii* was cultivated in tissue culture for six months, then transplanted to a wild-mimic cultivation site. The samples were harvested at different times to determine and compare their kinsenoside content.

### 3.7. Statistical Data Analysis

The yield of kinsenoside was calculated as follows:Yield of kinsenoside extraction (%) DW = [(Y-192050)/84,911]*V/W*100,(3)
where Y is the relative peak area, V is the total volume after extraction (mL) and W is the dry weight of the sample (g).

All experiments were done in triplicate. Statistical and mapping analyses were conducted by using Design Expert version 8.0.5.0 (Stat-ease INC., Minneapolis, MN, USA) and Microsoft Office Excel 2010 software (Microsoft Corporation, Redmond, WA, USA). One-way ANOVA was used to test all individual data groups to determine the significant differences using the IBM SPSS ver. 24 statistics software (IBM, Armonk, NY, USA). A Fisher’s least significant difference (LSD) multiple comparison test, at *p* < 0.05, was applied to determine differences among data groups.

## 4. Conclusions

The present study was an attempt to determine the ultrasonication factors and mechanisms optimizing kinsenoside extraction yield from *A. roxburghii* by using RSM as mathematical tool for extraction process optimization. The experimental results for the extraction of kinsenoside were found in close agreement with predicted ones. Statistical and graphical analysis showed that the interaction of ultrasonic temperature and methanol concentration has notable influence on the kinsenoside extraction yield. The liquid-solid ratio played the most important role in the extraction of kinsenoside compounds from *A. roxburghii*, probably due to the dissolving capability of the solvents. The optimal conditions required for high yields of kinsenoside (35.00 °C ultrasonic temperature, 16.33% methanol concentration, 10.83:1 liquid-solid ratio) were estimated using the model equation. Under this condition, kinsenoside extraction yield reached 32.24% DW. The optimized extraction process can get higher yield of kinsenoside. Furthermore, a determination and variation trend of the content of kinsenoside were analyzed for the samples at different cultivation ages. The present study could be further used to consider *A. roxburghii* for the extraction of kinsenoside under optimized extraction conditions and utilized in industrial suppliers of the food, pharmaceutical, and beverages products.

## Figures and Tables

**Figure 1 molecules-25-00193-f001:**
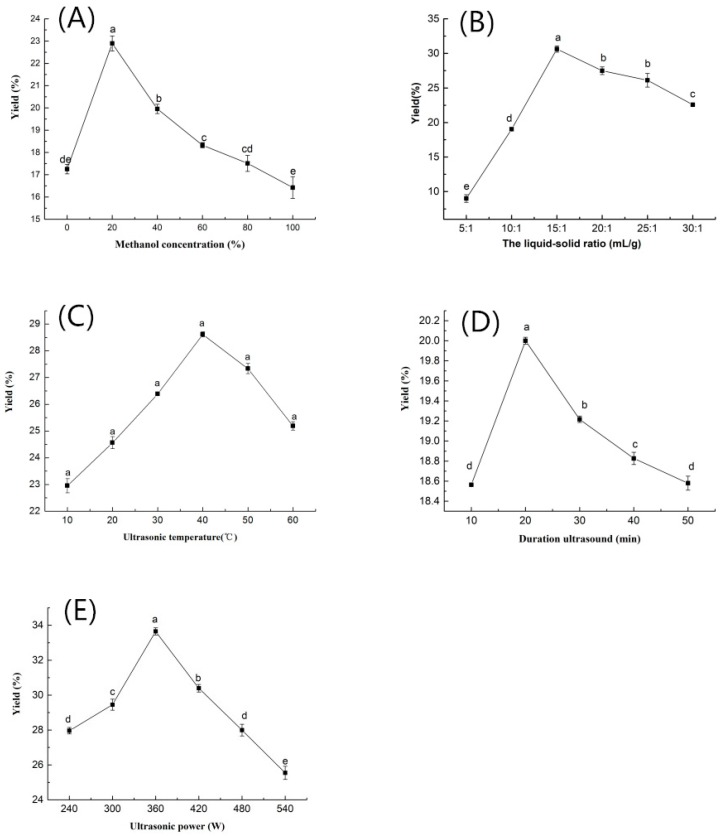
Effects of five factors on kinsenoside yield. (**A**) Effects of methanol concentration. (**B**) Effects of the liquid-solid ratio. (**C**) Effects of ultrasonic temperature. (**D**) Effects of the duration of ultrasound. (**E**) Effects of ultrasonic power. Note: Different letters in the figures indicate significant difference at 0.05 level.

**Figure 2 molecules-25-00193-f002:**
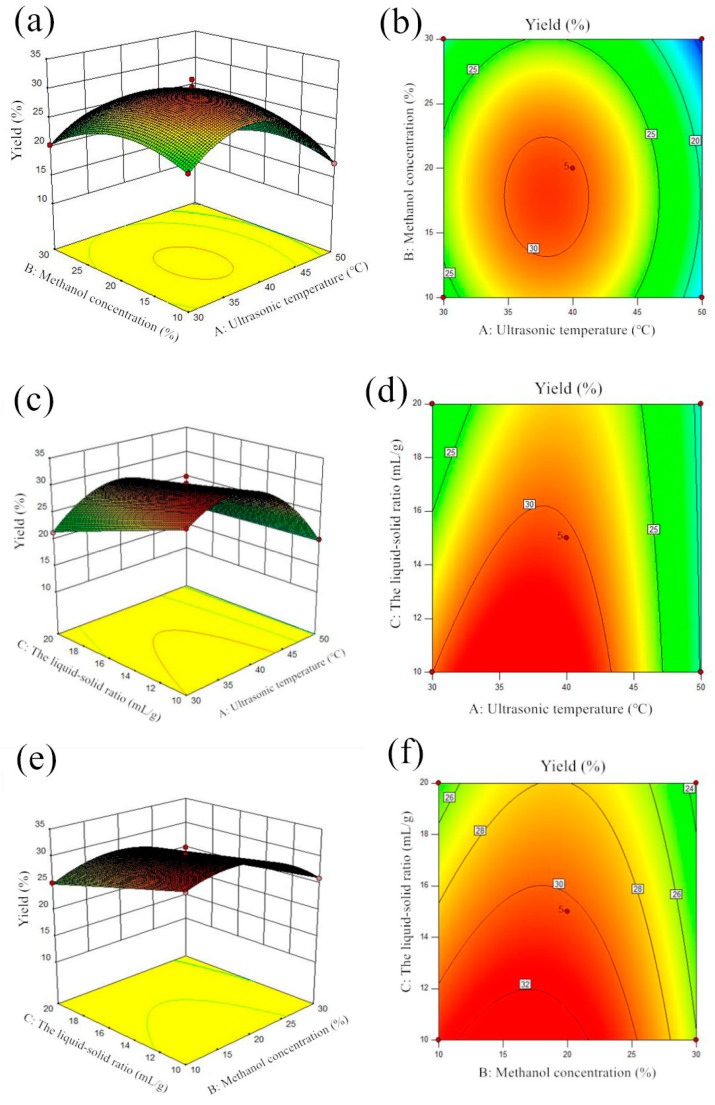
Contour map and response surface plot. (**a**,**b**) Interaction plot of ultrasonic temperature and methanol concentration (the liquid-solid ratio is constant at 15:1 mL/g). (**c**,**d**) Interaction plot of ultrasonic temperature and the liquid-solid ratio (the methanol concentration is constant at 20%). (**e**,**f**) Interaction plot of methanol concentration and the liquid-solid ratio (the ultrasonic temperature is constant at 40 °C).

**Figure 3 molecules-25-00193-f003:**
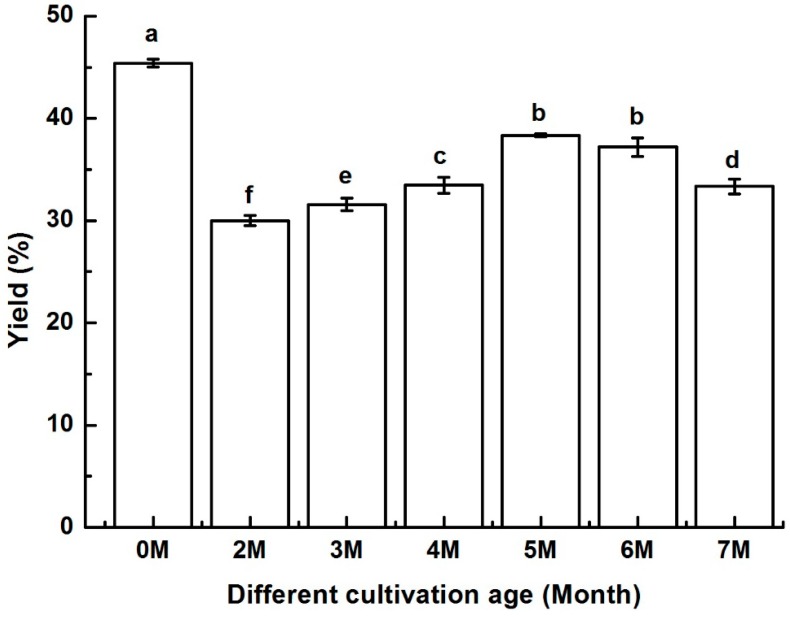
Kinsenoside yield in *A. roxburghii* at different cultivation ages. Note: Different letters in the figures indicate significant difference at 0.05 level.

**Figure 4 molecules-25-00193-f004:**
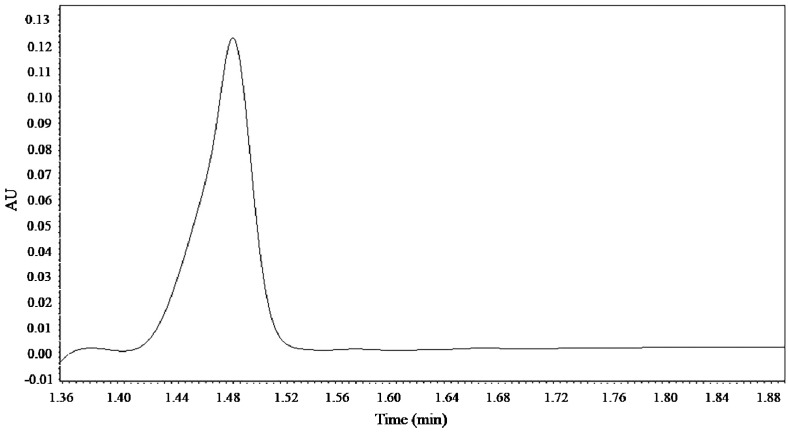
UPLC chromatograms of kinsenoside reference substances.

**Table 1 molecules-25-00193-t001:** ANOVA analysis of Box-Behnken test.

Source	Sum of Square	Degree of Freedom	Mean Square	F Value	*p*-Value	Significant
Model	482.13	9	53.57	117.59	<0.0001	**
A	77.5	1	77.50	170.12	<0.0001	**
B	22.71	1	22.71	49.86	0.0002	**
C	40.77	1	40.77	89.49	<0.0001	**
AB	0.027	1	0.027	0.06	0.8139	
AC	15.17	1	15.17	33.3	0.0007	**
BC	3.55	1	3.55	7.8	0.0268	*
A^2^	245.98	1	245.98	539.93	<0.0001	**
B^2^	60.74	1	60.74	133.34	<0.0001	**
C^2^	6.16 × 10^−3^	1	6.16 × 10^−3^	0.014	0.9107	
Residual	3.19	7	0.46			
Lack of fit	0.28	3	0.093	0.13	0.9382	
Error	2.91	4	0.73			R^2^ = 0.993
Total	485.32	16				Radj^2^ = 0.985

Note: ** in the table indicate significant difference at 0.01 level; * indicate significant difference at 0.05 level.

**Table 2 molecules-25-00193-t002:** Design of single factor experiment.

Independent Variable	Factor Level
Methanol concentration (%)	0, 20, 40, 60, 80, 100
The liquid-solid ratio (mL/g)	5:1, 10:1, 15:1, 20:1, 25:1, 30:1
Ultrasonic temperature (°C)	10, 20, 30, 40, 50, 60
Ultrasonic time (min)	10, 20, 30, 40, 50
Ultrasonic power (W)	240, 300, 360, 420, 480, 540
Extraction times (min)	1, 2, 3, 4

**Table 3 molecules-25-00193-t003:** Factors Level in RSM experiment.

Independent Variable	Factor Level		
	−1	0	1
Ultrasonic temperature (°C)	30	40	50
Methanol concentration (%)	10	20	30
The liquid-solid ratio (mL/g)	10:1	15:1	20:1

**Table 4 molecules-25-00193-t004:** Design and results of Box-Behnken test.

No.	Factor			Yield (% DW)
	A	B	C	
1	30	10	15	23.66
2	30	20	10	30.09
3	30	30	15	20.39
4	30	20	20	21.32
5	40	10	10	31.15
6	40	30	10	25.96
7	40	20	15	29.72
8	40	20	15	29.53
9	40	20	15	30.41
10	40	20	15	30.21
11	40	20	15	31.7
12	40	10	20	25.11
13	40	30	20	23.69
14	50	20	10	20.05
15	50	10	15	17.52
16	50	30	15	13.92
17	50	20	20	19.07

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
