# Peer review of "Optimization of Ultrasound Assisted Extraction (UAE) of Kinsenoside Compound from Anoectochilus roxburghii (Wall.) Lindl by Response Surface Methodology (RSM)"

_molecules, 2020, doi:10.3390/molecules25010193_

Round 1

Reviewer 1 Report

The paper deals with   Ultrasound Assisted Extraction (UAE) of Kinsenoside Compound from Anoectochilus Roxburghii (wall.) Lindl by Response Surface Methodology (RSM) . The article is well written and studied but the following aspects should be taken into account in order to improve the quality of the paper:

Authors have to UPDATE their references and also to show some original aspects of using ultrasound as “green” technologies and to detailed what are the green impacts : reduction of water, energy, wastes… These references could help the authors for the green impacts.

Chemat et al

Review of Green Food Processing techniques. Preservation, transformation, and extraction

Innovative Food Science and Emerging Technologies 41 (2017) 357–377 

 2. Previous articles claimed that ultrasound increases degradations of natural products. These articles have shown by chemical analysis that there is degradation when ultrasound is used for treatment of natural products. Authors have to discuss the effect of US on degradation metabolites on the degradation point of view. But authors have to be care of to separate the degradation due US and these due of water Allaf et al. DIC and Ultrasound for extraction of antioxidants and essential oils

ULTRASONICS SONOCHEMISTRY ; 20 (2013) 239-246

Authors have to propose a mechanism using their microscopic study to show ultrasound effects.

Author Response

The paper deals with  Ultrasound Assisted Extraction (UAE) of Kinsenoside Compound from Anoectochilus Roxburghii (wall.) Lindl by Response Surface Methodology (RSM) . The article is well written and studied but the following aspects should be taken into account in order to improve the quality of the paper:

Point 1: Authors have to UPDATE their references and also to show some original aspects of using ultrasound as “green” technologies and to detailed what are the green impacts : reduction of water, energy, wastes… These references could help the authors for the green impacts.

Chemat et al. Review of Green Food Processing techniques. Preservation, transformation, and extraction. Innovative Food Science and Emerging Technologies 41 (2017) 357–377.

Response 1: Thanks for your great suggestion. We have updated the references and described the ultrasound as “green” technologies and described its special benefits in the Introduction section.

Point 2: Previous articles claimed that ultrasound increases degradations of natural products. These articles have shown by chemical analysis that there is degradation when ultrasound is used for treatment of natural products. Authors have to discuss the effect of US on degradation metabolites on the degradation point of view. But authors have to be care of to separate the degradation due US and these due of water Allaf et al. DIC and Ultrasound for extraction of antioxidants and essential oils

ULTRASONICS SONOCHEMISTRY ; 20 (2013) 239-246

Authors have to propose a mechanism using their microscopic study to show ultrasound effects.

Response 2: Thanks for your great point. We have discussed the effect of US on degradation metabolites under the high ultrasonic temperature, the long-duration extraction and the high ultrasonic power in our revised manuscript. The further verification needs to apply SEM (Scanning Electron Microscope) to verify the cell structure to show the ultrasound effects. But now we don't have the materials for further verification. Therefore, we will do this in our future research.

Reviewer 2 Report

There are errors in the text very evident, font type different. References in text differents etc.

I consider that the manuscript must have more results that only the yield, for example, the kinsenoside quantity. 

I did an explanation in the revision, the analysis must be the quantification of kinsenoside.

Author Response

Point 1: There are errors in the text very evident, font type different. References in text differents etc.

Response 1: Thanks for your comments. We have thoroughly revised our manuscript and made our best to correct the possible errors.

Point 2: I consider that the manuscript must have more results that only the yield, for example, the kinsenoside quantity.

Response 2: Thanks for your comments. We have considered the effect of six single factors on the kinsenoside yield, including methanol concentration (0-100%), liquid-solid ratio (5:1-30:1 mL/g), ultrasonic power (240-540W), duration of ultrasound (10-50min), ultrasonic temperature (10-60℃) and the number of extractions (1-4). Moreover, we discussed the effects one by one in Results and Discussions sections. For example, the effect of extraction times on yield. Considering the cost factors of electricity and extraction time, we chose 2 times as a relatively good extraction factor.

Although the result showed that with the increase of the number of extractions, the kinsenoside also increases accordingly, we considered that 2 times of extraction the best and used in our experiments, since when the number of extractions is greater than 2, the small amount of the extract increased would not match the energy and time consumed during the process.

Point 3: I did an explanation in the revision, the analysis must be the quantification of kinsenoside.

Response 3: Thanks. But unfortunately, we haven’t seen your explanation. Nevertheless, we used the weight of the extracted kinsenoside in grams per gram of dry weight of Anoectochilus Roxburghii (wall.) Lindl (the percentage of the kinsenoside extracted) as the quantity of the kinsenoside, which we consider maybe slightly more suitable than the weight of the extracted kinsenoside itself.

Reviewer 3 Report

The study is of interest for the extraction of natural products, the interest is average, however I find a lack of the study of the influence in the extraction of individual components.

Author Response

Point 1: The study is of interest for the extraction of natural products, the interest is average, however I find a lack of the study of the influence in the extraction of individual components.

Response 1: Thanks for your comments. We have considered the effect of six single factors on the yield, including methanol concentration (0-100%), liquid-solid ratio (5:1-30:1 mL/g), ultrasonic power (240-540W), duration of ultrasound (10-50min), ultrasonic temperature (10-60℃) and the number of extractions (1-4). Moreover, we discussed the effects one by one in Results and Discussions section.

Reviewer 4 Report

The manuscript titled “Optimization of Ultrasound Assisted Extraction (UAE) of Kinsenoside Compound from Anoectochilus Roxburghii (wall.) Lindl by Response Surface Methodology (RSM)” deals with the identification of the best method to adopt for the extraction of the glycoside kinsenoside from the whole plant Anoectochilus roxburghii.

The manuscript was well written with a good rationale and the results are consistent and well discussed. Therefore I suggest its publication after minor revisions.

Please rewrite all the manuscript carefully following the authors guidelines. In particular the bibliography which doesn’t follow them in both text and chapter. Materials and Methods must be reported after the results and the discussions, whereas the Conclusions must be placed at the end of the article. Please double check the manuscript for typos. It is task of the authors to do it

Please try to follow the author guidelines and the manuscript can be accepted for pubblication

Author Response

Point 1: The manuscript titled “Optimization of Ultrasound Assisted Extraction (UAE) of Kinsenoside Compound from Anoectochilus Roxburghii (wall.) Lindl by Response Surface Methodology (RSM)” deals with the identification of the best method to adopt for the extraction of the glycoside kinsenoside from the whole plant Anoectochilus roxburghii.

Response 1: Thanks for the reviewer’s comments.

Point 2: The manuscript was well written with a good rationale and the results are consistent and well discussed. Therefore I suggest its publication after minor revisions.

Response 2: Thanks. We have done our best to revise our manuscript as required.

Point 3: Please rewrite all the manuscript carefully following the authors guidelines. In particular the bibliography which doesn’t follow them in both text and chapter. Materials and Methods must be reported after the results and the discussions, whereas the Conclusions must be placed at the end of the article. Please double check the manuscript for typos. It is task of the authors to do it. Please try to follow the author guidelines and the manuscript can be accepted for publication.

Response 3: Thanks. We have done our best to thoroughly revise our manuscript as required.

Reviewer 5 Report

The manuscript covers a very interesting aspect concerning the best conditions for Ultrasound Assisted Extraction of kinsenoside from Anoectochilus Roxburghii by response surface methodology

The manuscript is well structured and focused,therefore I would recommend its publication just with few minimal corrections:

- Introduction section (line 10): ZHANG, PENG, LI & SONG not in capital letter

- Introduction section (line 15): please erase etc after anti-osteoporosis

- Introduction section, page 2, line 21: please change soluble sugar, protein with soluble sugars and proteins; please erase and polyphenolics

- Introduction section, page 2, line 29: ailments is for aliments ?

Line 43: Please erase the word set (it is used twice in the same line)

-Section 2.2, line 2: please change the pure water with pure water; please change Hangzhou Wahaha group Co. Ltd. with Hangzhou Wahaha group Co. Ltd. (Hangzhou, China)

- Figures 1 and 2 have a very low resolution.

Author Response

Point 1: Introduction section (line 10): ZHANG, PENG, LI & SONG not in capital letter.

Response 1: Thanks. We have made modifications as required.

Point 2: Introduction section (line 15): please erase etc after anti-osteoporosis.

Response 2: Thanks. But as there are more functions discussed in the cited literature than those listed in this sentence, we think it would be better to keep etc after anti-osteoporosis.

Point 3: Introduction section, page 2, line 21: please change soluble sugar, protein with soluble sugars and proteins; please erase and polyphenolics.

Response 3: Thanks. We have changed soluble sugar, protein to soluble sugars and proteins. By the way, polyphenols was misspelt as polyphenolics in the original manuscript and we have changed it back to 'polyphenols' in the revised version. Sorry for our mistake.

Point 4: Introduction section, page 2, line 29: ailments is for aliments ?

Response 4: Thanks. What we were trying to mean was ailments.

Point 5: Line 43: Please erase the word set (it is used twice in the same line).

Response 5: Thanks. We have tried to correct all the possible errors. However, it seams that there is no such a word of ‘set’ in our manuscript.

Point 6: Section 2.2, line 2: please change the pure water with pure water; please change Hangzhou Wahaha group Co. Ltd. with Hangzhou Wahaha group Co. Ltd. (Hangzhou, China).

Response 6: Thanks. We have made modifications as required.

Point 7: Figures 1 and 2 have a very low resolution.

Response 7: Thanks. We have made provided figures with higher resolution.

Round 2

Reviewer 2 Report

The authors Could include the data of the standard curve of kinsenoside and a UPLC-spectrum that include the kinsenoside standard?. Also, it is important that in the manuscript include de mg/g DW of kinsenoside.

Author Response

Point 1: The authors Could include the data of the standard curve of kinsenoside and a UPLC-spectrum that include the kinsenoside standard?. Also, it is important that in the manuscript include de mg/g DW of kinsenoside.

Response 1:

Thanks for the suggestion. The standard curve of kinsenoside is a linear one and the regression equation of it was given in Subsection 3.2. A UPLC-spectrum that includes the kinsenoside standard is added to our manuscript. Thanks.

As for the unit of the yield of kinsenoside, it was expressed as %DW, that is, mg/mg*100% DW, which has the same meaning as mg/g DW with only one magnitude smaller than that in figure. If it were expressed as mg/g DW, all yields would be in the magnitude hundreds, which would be slightly more difficult to be understood. Therefore, we just keep it and did not change.

Many thanks again.
